# Head-wise Adaptive Rotary Positional Encoding for Fine-Grained Image Generation

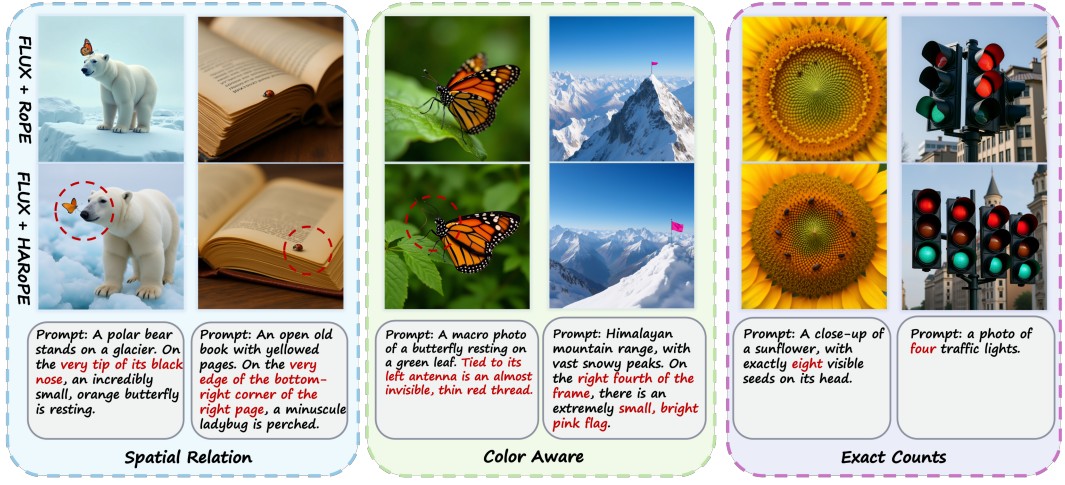

Figure 1: Qualitative comparison of generated images across three fine-grained challenges: spatial relations (left), color fidelity (middle), and object counting (right). HARoPE consistently outperforms RoPE, adhering more faithfully to prompt specifications (instruction keywords highlighted in red).

## ABSTRACT

Transformers rely on explicit positional encoding to model structure in data. While Rotary Position Embedding (RoPE) excels in 1D domains, its application to image generation reveals significant limitations such as fine-grained spatial relation modeling, color cues, and object counting. This paper identifies key limitations of standard multi-dimensional RoPE—rigid frequency allocation, axis-wise independence, and uniform head treatment—in capturing the complex structural biases required for fine-grained image generation. We propose HARoPE, a head-wise adaptive extension that inserts a learnable linear transformation parameterized via singular value decomposition (SVD) before the rotary mapping. This lightweight modification enables dynamic frequency reallocation, semantic alignment of rotary planes, and head-specific positional receptive fields while rigorously preserving RoPE's relative-position property. Extensive experiments on class-conditional ImageNet and text-to-image generation (Flux and MMDiT) demonstrate that HARoPE consistently improves performance over strong RoPE baselines and other extensions. The method serves as an effective drop-in replacement, offering a principled and adaptable solution for enhancing positional awareness in transformer-based image generative models.

## 1 INTRODUCTION

Transformers are inherently permutation-invariant and therefore require explicit positional signals to model order and structure in sequential and spatial data Vaswani et al. (2017). Positional embeddings meet this need by mapping position indices to vectors of the same dimensionality as token features, enabling the model to fuse positional and semantic information without architectural changes. Two

broad families are widely used. Absolute positional encodings assign a unique vector to each index, implemented either as fixed sinusoidal functions Vaswani et al. (2017); Chen et al. (2023); Peebles & Xie (2022) or as learned embeddings Gehring et al. (2017). Relative encodings instead inject pairwise offset information directly into the attention mechanism Shaw et al. (2018); Raffel et al. (2020); Dai et al. (2019), often improving structural bias and length generalization. Among these, Rotary Positional Embedding Su et al. (2024); Barbero et al. (2024); Su (2021); Liu et al. (2023) is particularly notable: it represents absolute positions as complex-plane rotations and induces attention scores that depend solely on relative offsets, yielding strong empirical performance and extrapolation-friendly behavior.

Despite its success in one-dimensional settings, RoPE faces fundamental challenges when extended to multi-dimensional data, especially in image generation, which requires fine-grained spatial relations, color-aware cues, and exact object counts (as shown in Figure 1). First, conventional designs partition feature dimensions uniformly across axes and reuse the same frequency spectrum, implicitly assuming comparable complexity, scale, and dynamics along each direction. This rigid allocation is often suboptimal, especially in heterogeneous domains where horizontal and vertical axes (or spatial and temporal dimensions) exhibit different frequency characteristics. Second, standard multi-dimensional constructions implement rotations on fixed, coordinate-indexed planes and enforce axis-wise independence through block-diagonal structures. These choices constrain positional encoding to predefined subspaces that may be misaligned with the model's learned semantics and suppress cross-dimensional interactions such as diagonal, rotational, or spatiotemporal couplings. Third, applying a single, shared positional mapping across all attention heads overlooks their distinct roles and receptive fields, limiting the emergence of head-level specialization needed to capture multi-scale and anisotropic patterns.

Motivated by these observations, we introduce HARoPE, a head-wise adaptive rotary positional encoding mechanism that preserves RoPE's relative-offset property while addressing the above limitations in a lightweight and modular manner. The key idea is to insert, immediately before the rotary mapping, a learnable linear transformation parameterized via a singular value decomposition (SVD). By projecting queries and keys through this SVD-based change of basis, HARoPE aligns rotary planes with semantically meaningful directions and facilitates explicit cross-axis mixing. Moreover, endowing each attention head with an independent SVD equips the model with specialized positional receptive fields, promoting complementary multi-scale behaviors. Crucially, using the same adaptation for queries and keys preserves RoPE's offset equivariance, encouraging that attention depends on positions only through relative differences.

Experiments on the ImageNet generation task demonstrate that HARoPE offers a simple, drop-in mechanism and obtains improved performance compared to naïve multi-dimensional RoPE and recent extensions. When integrated into text-to-image generative models (Flux and MMDiT), HARoPE yields consistent gains, indicating that adaptive, head-wise positional rebasing complements large-scale text-to-image generative architectures.

## 2 RELATED WORKS

**Position Embedding in Transformers.** Transformers are permutation-equivariant and therefore require positional signals to model order and structure Vaswani et al. (2017). Early approaches include learned absolute embeddings Chu et al. (2021); Gehring et al. (2017) and fixed sinusoidal encoding Vaswani et al. (2017); Chen et al. (2023); Peebles & Xie (2022), the latter enabling length extrapolation. Relative schemes Shaw et al. (2018); Raffel et al. (2020); Dai et al. (2019) inject pairwise distance information directly into attention, improving structural bias across diverse tasks.

**RoPE and its Extensions.** RoPE encodes absolute positions via complex-plane rotations while preserving a strict relative-offset property in attention Su et al. (2024). RoPE's parameter-free, extrapolative design has driven broad adoption in large language models. However, its original 1D formulation is not directly aligned with the multi-dimensional inputs common in vision. Several works extend RoPE beyond 1D: RoPE-ViT generalizes to images Heo et al. (2024), and MRoPE supports 2D/3D and multimodal settings Wang et al. (2024b); Bai et al. (2025). Despite progress, common designs (i) uniformly partition feature dimensions across axes, (ii) enforce axis-wise independence via block-diagonal rotations, and (iii) apply identical positional mappings across

heads—limitations that hinder alignment with learned semantics, cross-axis coupling, and head specialization. Complementary efforts provide broader foundations: RethinkRoPE Liu et al. (2025) offers a systematic mathematical blueprint for higher-dimensional RoPE, and STRING Schenck et al. (2025) introduces learnable matrix generalizations. Building on these insights, we study the learnable-matrix setting and introduce a lightweight, head-specific linear adaptation via SVD parameterization that preserves RoPE's relative-offset property while enabling semantic alignment, cross-axis mixing, and per-head specialization.

**Image Generation and Understanding.** Diffusion-based text-to-image systems (e.g., DALL·E Ramesh et al. (2021)), DiT Peebles & Xie (2022), Stable Diffusion Rombach et al. (2022), Flux Labs et al. (2025)) achieve state-of-the-art generation by coupling strong text encoders with scalable Transformers. In visual understanding, ViT Dosovitskiy et al. (2020); Heo et al. (2021); Beyer et al. (2023); Li et al. (2024) and Swin Transformer Liu et al. (2021) have largely supplanted convolutional backbones by modeling long-range dependencies and enabling multimodal alignment. In both generation and understanding tasks, effective positional encoding is critical for representing spatial and spatiotemporal structure. The proposed HARoPE method is complementary to these Transformer-based approaches. We demonstrate its efficacy in image generation using Flux and MMDiT, and in image understanding with ViT-Base.

## 3 METHODOLOGY

We introduce HARoPE (Head-wise Adaptive Rotary Positional Encoding), a drop-in enhancement to RoPE designed to preserve its desirable relative-position property while addressing three core limitations that arise in multi-dimensional settings: rigid frequency allocation, misalignment with learned semantic subspaces, and uniform treatment across attention heads. HARoPE incorporates a lightweight, head-specific linear transformation—parameterized via a singular value decomposition—immediately before the rotary mapping. This adaptation enables (i) dynamic redistribution of positional capacity across axes, (ii) semantic alignment of rotary planes and support for cross-axis interactions, and (iii) specialized positional receptive fields per attention head.

We first review the original RoPE and a common multi-dimensional extension (Section 3.1), then detail the specific limitations of the standard approach (Section 3.2), and finally present the HARoPE formulation and its properties (Section 3.3).

### 3.1 PRELIMINARY: ROTARY POSITION EMBEDDINGS

**One-Dimensional RoPE.** RoPE injects position via 2D rotations applied to consecutive feature pairs. For a feature vector $\mathbf{q} \in \mathbb{R}^d$ at position $m$, define the block-diagonal rotation

$$R_m = \mathrm{diag}\bigg( \begin{bmatrix} \cos(m\theta_0) & -\sin(m\theta_0) \\ \sin(m\theta_0) & \cos(m\theta_0) \end{bmatrix}, \ldots, \begin{bmatrix} \cos(m\theta_{d/2-1}) & -\sin(m\theta_{d/2-1}) \\ \sin(m\theta_{d/2-1}) & \cos(m\theta_{d/2-1}) \end{bmatrix} \bigg), \quad (1)$$

with frequencies $\theta_i = \theta_{\mathrm{base}}^{-2i/d}$ (typically $\theta_{\mathrm{base}} = 10000$). Rotated queries and keys are $\mathbf{q}' = R_m\mathbf{q}$, $\mathbf{k}' = R_n\mathbf{k}$. A key property is relative-position encoding:

$$(R_m\mathbf{q})^\top (R_n\mathbf{k}) = \mathbf{q}^\top R_{n-m}\mathbf{k}, \quad (2)$$

so attention scores depend on the offset $n - m$ only. Each pair $(q_{2i}, q_{2i+1})$ forms a 2D plane rotated by phase $m\theta_i$, yielding a multi-frequency spectrum.

**A Naïve Multi-dimensional Extension.** For 2D positions $(x, y)$, a standard extension partitions the feature dimensions across axes and applies independent rotations:

$$R_{(x,y)} = \mathrm{diag}\big(R_x(x), R_y(y)\big), \quad (3)$$

where $R_x(\cdot)$ and $R_y(\cdot)$ reuse the 1D spectrum. With $\mathbf{q} = [\mathbf{q}_x; \mathbf{q}_y]$, $\mathbf{k} = [\mathbf{k}_x; \mathbf{k}_y]$, the rotated vectors and the score can be written as

$$\mathbf{q}' = \begin{bmatrix} R_x(x) & 0 \\ 0 & R_y(y) \end{bmatrix} \mathbf{q}, \quad \mathbf{k}' = \begin{bmatrix} R_x(x') & 0 \\ 0 & R_y(y') \end{bmatrix} \mathbf{k}, \quad (4)$$

$$\mathbf{q'}^{\top}\mathbf{k'} \;=\; \underbrace{\mathbf{q}_x^{\top} R_x(x)^{\top} R_x(x')\mathbf{k}_x}_{x\text{-axis}} \;+\; \underbrace{\mathbf{q}_y^{\top} R_y(y)^{\top} R_y(y')\mathbf{k}_y}_{y\text{-axis}}. \tag{5}$$

This separability extends to higher dimensions by adding more axis-specific blocks.

### 3.2 Limitations of Naïve Multi-Dimensional RoPE

**Rigid Frequency Allocation.** Features are split evenly across axes and each axis reuses the same spectrum $\theta_i = 10000^{-2i/d}$, where the $\theta_{base}$ is manually predefined, implicitly assuming equal complexity and scale across directions. This assumption is often violated (e.g., temporal vs. spatial variation), leading to suboptimal capacity and frequency coverage.

**Semantic Misalignment and Axis Independence.** Rotations act on fixed, coordinate-indexed planes $(q_0, q_1), (q_2, q_3), \ldots$, irrespective of the semantic subspaces learned by the model. The block-diagonal structure further enforces axis-wise independence, suppressing explicit cross-axis interactions (e.g., diagonal or rotational couplings).

**Head-Wise Uniformity.** Standard RoPE injects the same positional mapping into every head, despite evidence that heads specialize in different receptive fields (local vs. long-range). This uniformity weakens multi-scale, head-specific positional sensitivity.

### 3.3 Head-Wise Adaptive RoPE

We propose HARoPE, a head-wise linear adaptation inserted immediately before the rotary mapping. The adaptation learns a change of basis that (i) reallocates positional capacity across axes, (ii) aligns rotary planes with semantically meaningful directions and enables cross-axis coupling, and (iii) allows different attention heads to specialize in distinct positional receptive fields—all while preserving RoPE's desirable relative-position property.

**Head-specific Linear Adaptation.** HARoPE inserts, for each attention head $h$ with per-head dimension $d$, a learnable linear transform $A_h \in \mathbb{R}^{d \times d}$ immediately before the rotary map. We parameterize

$$A_h = U_h \Sigma_h V_h^{\top}, \tag{6}$$

where $U_h, V_h$ are orthogonal and $\Sigma_h$ is diagonal with positive entries. Queries and keys at positions $m$ and $n$ are mapped as

$$\mathbf{q}_h' = R_m A_h \mathbf{q}_h, \qquad \mathbf{k}_h' = R_n A_h \mathbf{k}_h. \tag{7}$$

This single linear step separates concerns: $V_h$ selects and mixes directions (aligning rotary planes with learned semantics), $\Sigma_h$ redistributes effective capacity by reweighting subspaces, and $U_h$ maps enriched signals back to the model's native basis. Initializing $A_h = I$ recovers the baseline at step zero, and keeping singular values near 1 preserves scale.

The same $A_h$ is applied to queries and keys, and position dependence remains confined to the rotary maps, HARoPE preserves strict relative-offset dependence:

$$(\mathbf{q}_h')^{\top}\mathbf{k}_h' \;=\; (R_m A_h \mathbf{q}_h)^{\top}(R_n A_h \mathbf{k}_h) \;=\; \mathbf{q}_h^{\top} A_h^{\top} R_{n-m} A_h \mathbf{k}_h. \tag{8}$$

So attention scores depend on positions only through the relative offset $n - m$.

**Multi-Dimensional Extension.** For positions $(x_1, \ldots, x_p)$ in $p$ dimensions, let $R_{(x_1, \ldots, x_p)}$ be the block-diagonal rotary map formed by axis-wise rotations. Applying the same head-specific adaptation,

$$\mathbf{q}_h' \;=\; R_{(x_1, \ldots, x_p)} A_h \mathbf{q}_h, \quad \mathbf{k}_h' \;=\; R_{(x_1', \ldots, x_p')} A_h \mathbf{k}_h, \tag{9}$$

yields the score

$$(\mathbf{q}_h')^{\top}\mathbf{k}_h' \;=\; \mathbf{q}_h^{\top} A_h^{\top} R_{(\Delta x_1, \ldots, \Delta x_p)} A_h \mathbf{k}_h, \tag{10}$$

with $\Delta x_i = x_i' - x_i$. Hence, HARoPE preserves relative encoding in multi-dimensional settings while allowing learned cross-axis mixing through the dense $A_h$.

**Initialization and stability.** To ensure compatibility with pretrained models and stable optimization, we initialize $A_h = I$ via $U_h = V_h = I$ and $\Sigma_h = I$. Orthogonality of $U_h$ and $V_h$ can be maintained by parameterizing them through the matrix exponential of skew-symmetric matrices. The diagonal of $\Sigma_h$ is kept positive by softplus and regularized to remain near one to avoid exploding/vanishing norms and to preserve the variance of queries and keys.

**Discussion.** HARoPE can be interpreted as learning a head-specific harmonic coordinate system: $V_h$ aligns rotary planes with semantically meaningful directions; $\Sigma_h$ modulates the effective frequency budget across these directions; and $U_h$ reintegrates the positionally enriched features. By allowing each head to specialize its positional receptive field, HARoPE overcomes the limitations of rigid frequency allocation, axis-wise independence, and head-wise uniformity, while rigorously preserving RoPE's relative-position equivariance.

# 4 EXPERIMENTS

This section evaluates HARoPE across image understanding, class-conditional image generation, and text-to-image generation. We first describe the experimental protocol (architectures, datasets, baselines, and metrics), then present comparative results followed by ablations, limitations and future work discussion.

## 4.1 EXPERIMENTAL SETUPS

**Implementation.** We adopt standard backbones and training strategies for each task. For image understanding, we train ViT-B from scratch with AdamW, learning rate $5 \times 10^{-4}$ and a 5-epoch warmup from $1 \times 10^{-6}$, batch size 256, and 300 training epochs. For class-conditional image generation, we use DiT-B/2 with a constant learning rate $1 \times 10^{-4}$, no weight decay, batch size 256, and EMA with decay 0.9999 for evaluation. For text-to-image generation, we fine-tune the pretrained FLUX.1-dev model for 4,000 iterations using LoRA (rank 32), AdamW with learning rate $2 \times 10^{-5}$, weight decay 0.01, and batch size 64. All training and testing are performed on a server of eight NVIDIA H100 GPUs.

**Dataset.** Datasets follow conventional practice. Image understanding experiments use ImageNet at $224 \times 224$ with standard resize and center-crop. For ImageNet generation, we encode images using Stable Diffusion's VAE into $z \in \mathbb{R}^{H/8 \times W/8 \times 4}$ with $H \in \{128, 256, 512\}$. Text-to-image experiments with FLUX model use the BLIP30-60k instruction-tuning set of 60k prompt–image pairs. For MMDiT-based text-to-image generation, we utilize the train split of the MS-COCO dataset Lin et al. (2014).

**Baselines.** We compare against strong positional encoding baselines. For image understanding, we include absolute positional embeddings (APE), 2D-RoPE in axial and mixed forms (Heo et al., 2024), STRING (Schenck et al., 2025)/Rethinking RoPE (Liu et al., 2025), and HARoPE. For class-conditional generation on ImageNet, we evaluate APE, Vanilla RoPE, 2D-RoPE (Axial), VideoRoPE (Wei et al., 2025), STRING/Rethinking RoPE, and HARoPE. For text-to-image generation, we directly replace RoPE in FLUX with HARoPE and APE in MMDiT for a controlled comparison.

**Metrics.** For image understanding, we report Top-1 accuracy. In class-conditional generation, we adopt ADM's TensorFlow evaluation suite Dhariwal & Nichol (2021) to report FID-50K (Heusel et al., 2017), Inception Score (Salimans et al., 2016), and Precision/Recall (Davis & Goadrich, 2006). For text-to-image generation, we employ GenEval (Ghosh et al., 2023) and DPG-Bench (Hu et al., 2024) for comprehensive assessment.

## 4.2 COMPARISON TO EXISTED WORKS

**Image Understanding.** Table 1 summarizes ViT-B results trained for 300 epochs. HARoPE achieves the best Top-1 accuracy of 82.76%, improving upon APE by 2.19% and surpassing the strongest RoPE variant (2D-RoPE Mixed at 81.51%). These gains indicate that head-wise adaptive rotary rebasing applies well to the image understanding task and is also compatible with visual

| Position Embedding | Steps | Top-1 Acc (224×224) |
|---|---|---|
| APE (Default) | 300 epoches | 80.57 |
| 2D-RoPE (Axial) | 300 epoches | 81.35 |
| 2D-RoPE (Mixed) | 300 epoches | 81.51 |
| STRING/Rethinking RoPE | 300 epoches | 80.96 |
| HARoPE | 300 epoches | **82.76** |

Table 1: Image understanding with different position embeddings on ViT-Base.

| Position Embedding | FID-50k↓ | IS↑ | Precision↑ | Recall↑ |
|---|---|---|---|---|
| APE (Default) | 11.47 | 110.04 | 0.72 | 0.54 |
| Vanilla RoPE | 9.81 | 121.75 | 0.73 | 0.53 |
| 2D-RoPE (Axial) | 9.49 | 124.78 | 0.74 | 0.54 |
| VideoRoPE | 10.86 | 118.84 | 0.71 | 0.54 |
| STRING/Rethinking RoPE | 9.31 | 125.09 | 0.74 | 0.54 |
| HARoPE | **8.90** | **127.01** | **0.74** | **0.55** |

Table 2: Image generation on ImageNet with different position embeddings on DiT-B/2, 1M steps.

| Method | Testing FLOPs | GenEval ↑ | DPG Bench ↑ | FID ↓ |
|---|---|---|---|---|
| *256 resolution* | | | | |
| MMDiT (APE) | 307G | – | – | 6.34 |
| MMDiT (HARoPE) | 309G | – | – | **5.22** |
| *1024 resolution* | | | | |
| FLUX (RoPE) | 5T | 0.7567 | 83.26 | – |
| FLUX (HARoPE) | 5T | **0.7710** | **83.77** | – |

Table 3: Performance of HARoPE applied to FLUX.1-dev and MMDiT on 1024 and 256 resolution.

understanding. However, in this paper, we focus on the task of fine-grained image generation, so we do not conduct extensive experimental analysis for visual understanding.

**Class-Conditioned ImageNet Generation.**    On ImageNet with DiT-B/2 (Table 2), HARoPE attains the lowest FID-50k (8.90) and the highest IS (127.01), while matching the strongest Precision (0.74) and achieving the best Recall (0.55). These results reflect improved fidelity and perceptual quality without sacrificing diversity compared to axis-separable or fixed-spectrum designs.

**Text-to-Image Generation.**    Replacing RoPE with HARoPE in FLUX yields consistent improvements on both GenEval and DPG-Bench (Table 3). On GenEval, the overall score increases from 0.7567 to 0.7710, while on DPG-Bench it improves from 83.26 to 83.77. The relative gain is more pronounced on GenEval, which emphasizes fine-grained compositional attributes (e.g., object counting, colors, and spatial relations), aligning with HARoPE's head-wise adaptive design that enhances spatial discrimination. Applying HARoPE to MMDiT further reduces FID from 6.34 (APE) to 5.22 (HARoPE), indicating improved fidelity. Qualitative comparisons are provided in Figure 2.

### 4.3 ABLATION STUDY

**Different Matrix Parameterizations.**    We conduct an ablation study to evaluate the impact of the matrix parameterization in HARoPE's adaptation module. As shown in Table 4, we compare three matrix types: normal matrices (without orthogonality constraints), orthogonal matrices, and our SVD-based parameterization. The baseline RoPE achieves an FID-50k of 9.49. Introducing a single normal matrix improves FID to 9.28, while orthogonal and SVD parameterizations yield 9.31 and 8.93 respectively, demonstrating that constrained matrix structures provide more stable optimization.

**Head-wise Specialization.**    As shown in Table 4, we extend each matrix type to be head-specific, and observe consistent improvements across all matrix parameterizations. The normal matrix with multi-head configuration reduces FID to 9.03, while the orthogonal matrix variant achieves 8.97. Our proposed HARoPE (RoPE + SVD + multi-head) obtains the best performance with FID-50k of 8.90 and IS of 127.01. This trend is corroborated in text-to-image generation with the FLUX model

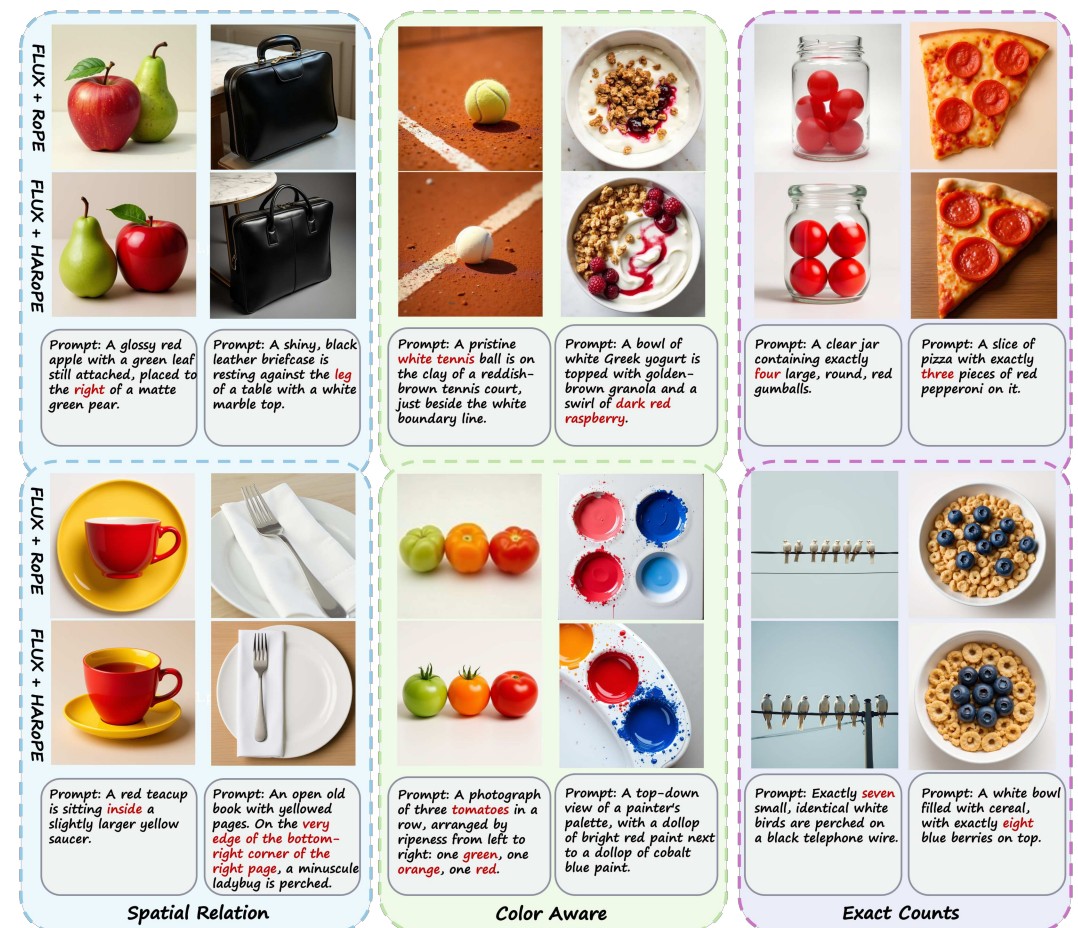

Figure 2: Qualitative comparison on wild prompts, evaluating FLUX models with RoPE and HARoPE positional embeddings.

| Position Embedding | Training steps | FID-50k↓ | IS↑ | Precision↑ | Recall↑ |
|---|---|---|---|---|---|
| RoPE | DiT-B/2,1M | 9.49 | 124.78 | 0.74 | 0.54 |
| RoPE + normal-matrix | DiT-B/2,1M | 9.28 | 124.78 | 0.74 | 0.53 |
| RoPE + normal-matrix + multi-head | DiT-B/2,1M | 9.03 | 126.89 | 0.74 | 0.53 |
| RoPE + orthogonal-matrix | DiT-B/2,1M | 9.31 | 125.09 | 0.74 | 0.54 |
| RoPE + orthogonal-matrix + multi-head | DiT-B/2,1M | 8.97 | 127.61 | 0.74 | 0.53 |
| RoPE + SVD | DiT-B/2,1M | 8.93 | 126.03 | 0.74 | 0.54 |
| RoPE + SVD + multi-head (Ours) | DiT-B/2,1M | **8.90** | **127.01** | 0.74 | **0.55** |

Table 4: Quantitative comparison of different matrix settings (normal, orthogonal and SVD parameterization; with and without multi-head separate learnable matrix) on ImageNet generation task.

(Table 9), where the head-wise variant yields superior scores on both GenEval. To further validate this specialization, we visualize the model weight of learned transformation matrices across different attention heads and transformer blocks in Figure 4. The distinct patterns observed in the heatmaps provide empirical evidence that different heads indeed learn divergent projection strategies, aligning with the intended design of head-wise adaptive positional encoding.

**Different Image Resolution.** We evaluate the robustness of HARoPE across multiple image resolutions to assess its scalability. As summarized in Table 5, HARoPE is applied to DiT-B/2 models trained for class-conditional generation at resolutions of $128 \times 128$, $256 \times 256$, and $512 \times 512$. The

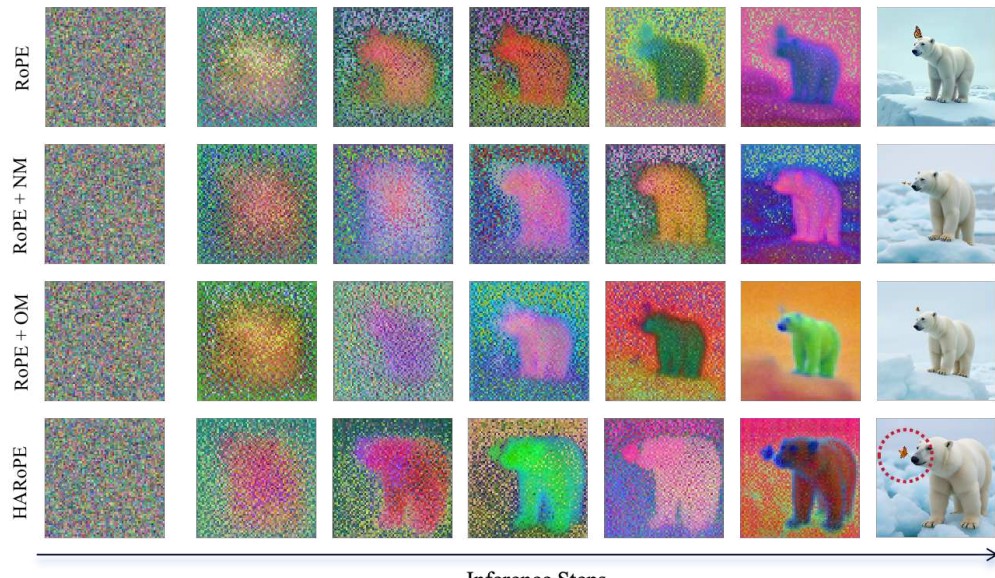

Prompt : A polar bear stands on a glacier . On the very tip of its black nose , an incredibly small,orange butterfly is resting .

Figure 3: Qualitative comparison of different matrix settings. During the inference steps, we demonstrate the "NM" denotes normal matrix, and "OM" denotes orthogonal matrix.

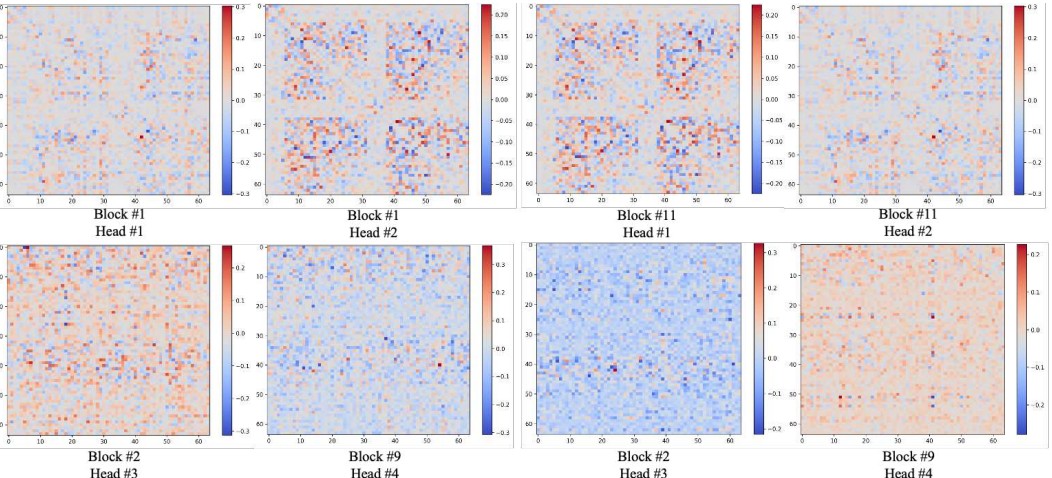

Figure 4: Model weight in heatmap of different learned matrices in different attention heads and different blocks.

results demonstrate that HARoPE consistently outperforms both Absolute Positional Embeddings and the standard RoPE baseline across all resolutions, achieving the best FID and IS scores. Furthermore, as shown in Table 3, when integrated into the large-scale FLUX model for text-to-image generation at a high resolution of $1024 \times 1024$, HARoPE again yields improved performance on both the GenEval and DPG-Bench metrics compared to the original RoPE.

**Extrapolation.** To assess the robustness of positional encodings, we evaluate their extrapolation capability—the ability to handle resolutions unseen during training. Models are trained on the standard ImageNet-1k resolution of $224 \times 224$ and tested at progressively larger resolutions. As shown in Table 6, HARoPE consistently achieves the highest accuracy across all evaluated resolutions.

| training steps | RoPE + SVD | RoPE + SVD + Multi-head (ours) |
|---|---|---|
| 500 | 0.7234 | 0.7292 |
| 1000 | 0.7206 | 0.7388 |

Figure 5: Comparing the performance of RoPE + SVD and RoPE + SVD + Multi-head on GenEval benchmark, FLUX model, 1024×1024 resolution.

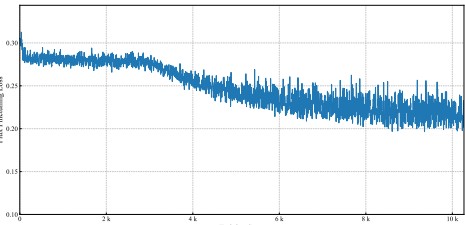

Figure 6: Training loss of the FLUX model during finetuning, showing stable and progressive convergence with HARoPE.

| Model | Position Embedding | FID-50k↓ | IS↑ | Precision↑ | Recall↑ |
|---|---|---|---|---|---|
| DiT-B/2, 128×128 | Absolution Embedding | 16.43 | 58.30 | 0.61 | 0.56 |
| DiT-B/2, 128×128 | RoPE | 14.32 | 66.13 | 0.62 | 0.57 |
| DiT-B/2, 128×128 | HARoPE | **13.73** | **68.14** | **0.63** | **0.57** |
| DiT-B/2, 256×256 | Absolution Embedding | 11.47 | 110.04 | 0.72 | 0.54 |
| DiT-B/2, 256×256 | RoPE | 9.49 | 124.78 | 0.74 | 0.54 |
| DiT-B/2, 256×256 | HARoPE | **8.90** | **127.01** | 0.74 | **0.55** |
| DiT-B/2, 512×512 | Absolution Embedding | 18.28 | 81.62 | 0.77 | 0.53 |
| DiT-B/2, 512×512 | RoPE | 14.57 | 95.41 | 0.79 | 0.53 |
| DiT-B/2, 512×512 | HARoPE | **14.36** | **96.25** | **0.80** | **0.54** |

Table 5: Image Generation Results on different image resolutions with our proposed HARoPE.

| Position Embedding | Model | step | 192 × 192 | 224× 224 | 256× 256 | 320 ×320 | 384 ×384 | 512× 512 |
|---|---|---|---|---|---|---|---|---|
| Original ViT (APE) | ViT-B | 300 epoch | 79.80 | 80.57 | 81.20 | 81.05 | 80.21 | 77.41 |
| 2D-RoPE (Axial) | ViT-B | 300 epoch | 80.39 | 81.35 | 82.00 | 82.34 | 81.93 | 80.11 |
| 2D-RoPE (Mixed) | ViT-B | 300 epoch | 80.50 | 81.51 | 82.22 | 82.62 | 82.12 | 80.63 |
| STRING/Rethinking RoPE | ViT-B | 300 epoch | 79.90 | 80.96 | 81.60 | 81.85 | 81.58 | 79.97 |
| HARoPE | ViT-B | 300 epoch | **81.75** | **82.76** | **83.36** | **83.92** | **83.70** | **82.88** |

Table 6: Extrapolation results of different position embedding methods in image understanding task.

Notably, at the extreme extrapolation size of $512 \times 512$, HARoPE maintains a strong accuracy of 82.88%, significantly outperforming other positional encoding methods.

**Efficiency and Training Stability.** As shown in Table 3, the TFLOPS introduced by the learnable matrices of HARoPE during inference can be very small compared to the entire model. The training process remains stable, as evidenced by the smooth and convergent loss curves during the fine-tuning of large models like FLUX (Figure 6).

## 5 CONCLUSION

Standard multi-dimensional extensions of RoPE face limitations in handling complex data like images, due to their rigid axis-wise feature partitioning, fixed rotation planes misaligned with semantic subspaces, and uniform application across attention heads. To overcome these issues, we introduced HARoPE, a head-wise adaptive rotary positional encoding that enhances RoPE through a lightweight, learnable linear transformation applied before the rotary mapping. Parameterized via singular value decomposition, this adaptation enables dynamic redistribution of positional capacity, semantic alignment of rotary planes with support for cross-axis interactions, and specialized positional receptive fields per attention head—all while preserving RoPE's strict relative-position encoding property. Extensive experiments on image understanding, class-conditional generation, and text-to-image synthesis demonstrate that HARoPE consistently outperforms existing positional encoding methods, confirming its effectiveness as a drop-in improvement for transformer-based generative models. These results highlight the value of adaptive, head-wise positional reasoning in capturing fine-grained structural and semantic patterns image generative models.

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

# A APPENDIX

## A.1 LIMITATIONS AND FUTURE WORKS

While HARoPE demonstrates consistent improvements across image understanding and generation tasks, this work has certain limitations that merit discussion. Our evaluation is primarily confined to the image domain due to our computational constraints; the generalizability of the approach to other multi-dimensional data modalities, such as video, audio, or 3D content, remains an open question for empirical validation.

Another consideration is the static nature of the learned transformation matrices, which are fixed after training. Although the head-wise specialization is beneficial, the adaptation process is not input-conditional. Exploring dynamic transformations that can adapt based on input content or evolve during inference could further enhance the flexibility and performance of the positional encoding mechanism.

## A.2 ADDITIONAL QUALITATIVE RESULTS

We provide supplementary visual comparisons to illustrate the empirical effects of HARoPE:

Figure 7 shows qualitative comparisons on GenEval prompts using FLUX with RoPE vs. HARoPE, highlighting improvements in spatial relations, color fidelity, and object counts. Figure 8 demonstrates text-to-image examples on MS-COCO using MMDiT, comparing APE/RoPE baselines and HARoPE. The results illustrate gains in fidelity and compositional consistency. Figure 9 visualizes HARoPE with and without head-wise specialization in FLUX.

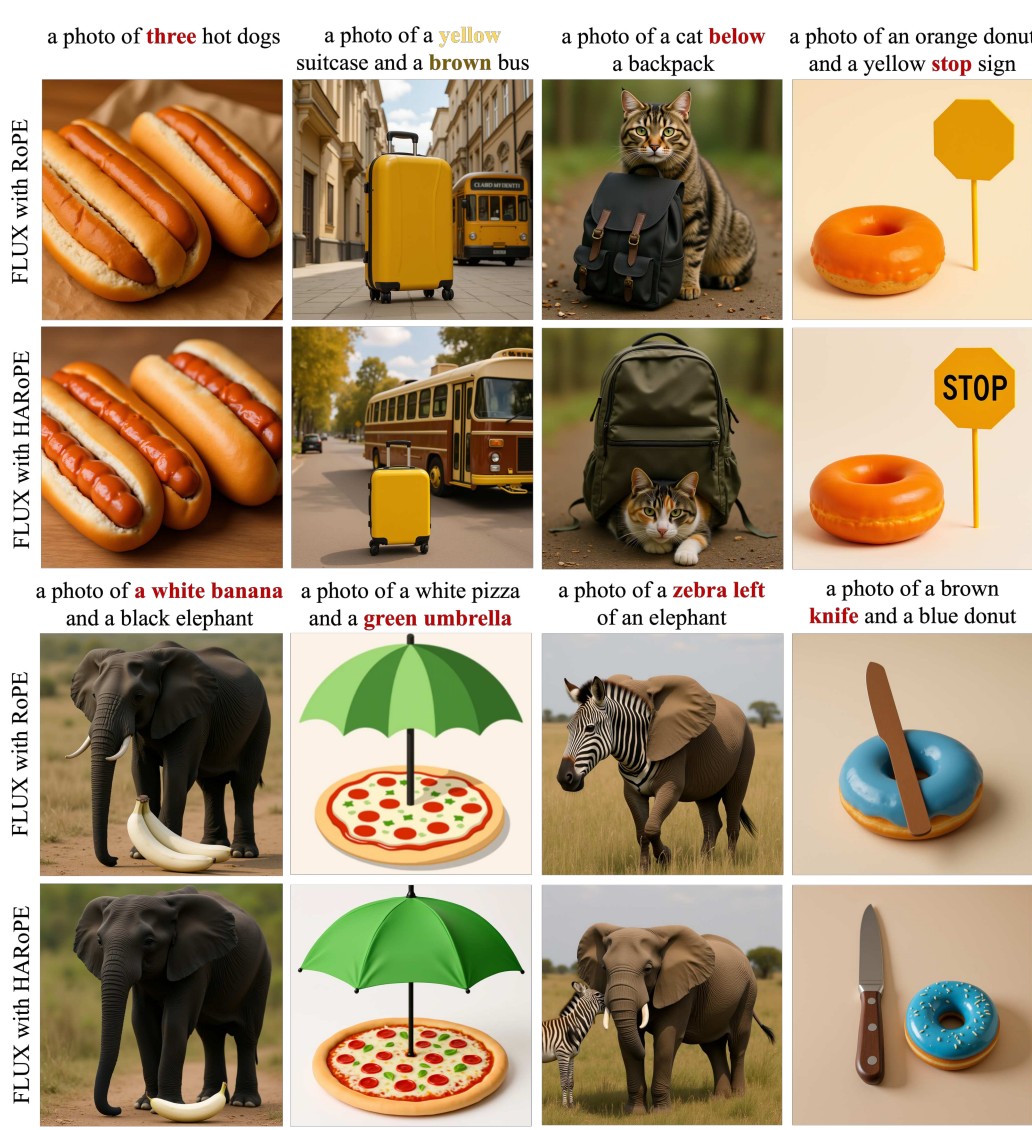

Figure 7: Qualitative Comparison on the GenEval Benchmark, evaluating FLUX models with RoPE and HARoPE positional embeddings.

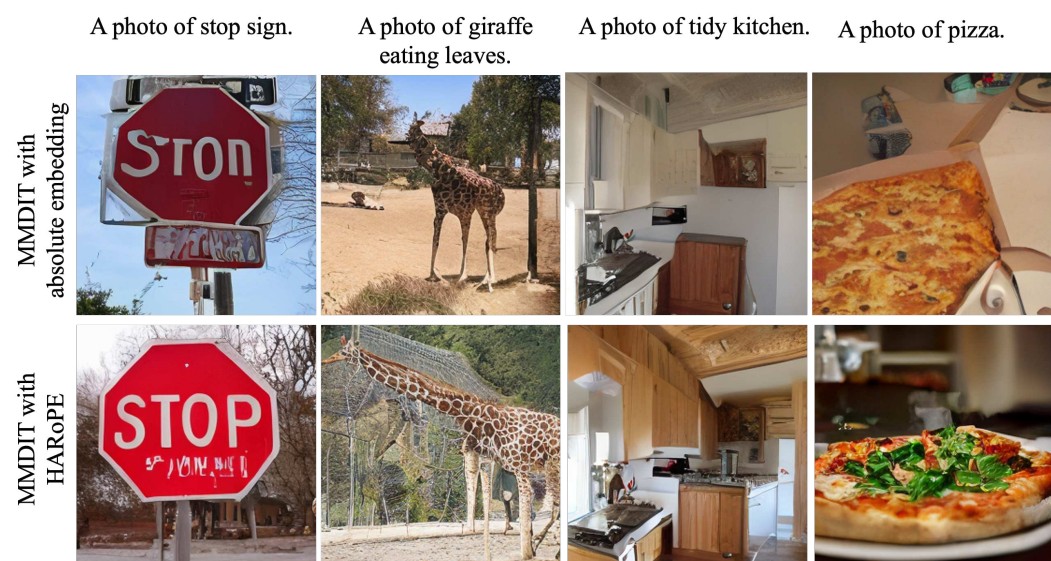

Figure 8: Text-to-image generation on MS-COCO. evaluating MMDiT models with RoPE and HARoPE positional embeddings.

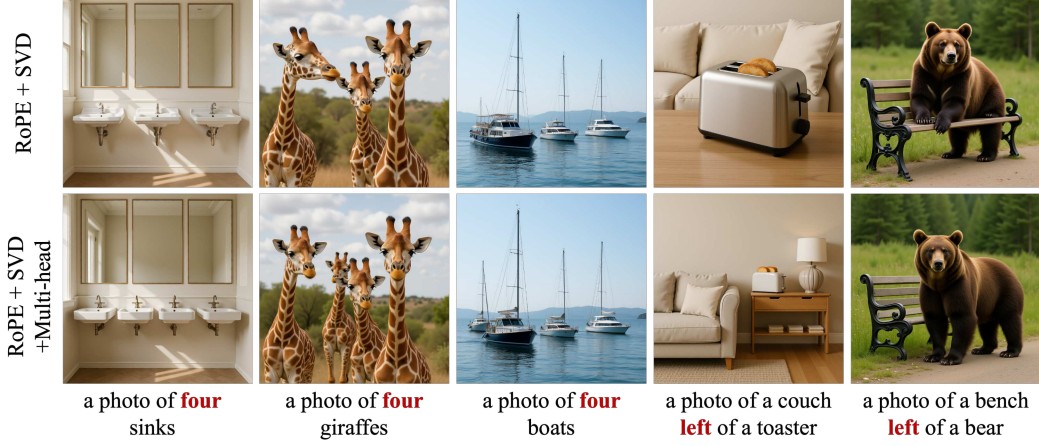

Figure 9: Visualization comparison of HARoPE with and with head-wise specialization, tested using Flux on the text-to-image generation task.

