# OpenReview forum: "Head-wise Adaptive Rotary Positional Encoding for Fine-Grained Image Generation"
_ICLR.cc/2026/Conference — ICLR 2026 Conference Withdrawn Submission_

### Official Review · Reviewer_LZZx · 2025-10-31

**Soundness:** 2
**Presentation:** 3
**Contribution:** 2
**Rating:** 4
**Confidence:** 3

**Summary:**

This paper aims at improving the rotary embedding in transformer achitectures. The method introduces a lightweight, head-specific learnable linear transformation that is applied to queries and keys before the standard rotary mapping. This transformation is parameterized via Singular Value Decomposition SVD. The authors conduct experiments on image understanding using ViT, image generation using DiT, and text-to-image generation wo validate the effectiveness.

**Strengths:**

- The paper writing is clear, and the method is simple and easy to follow.
- The experiments covers multiple aspects, i.e., image understanding, image generation.
- The ablations is detailed and clear.

**Weaknesses:**

- The method seems engineered and lacks novelty.
- The paper describes the method as lightweight. However, it introduces $N_{\text{heads}} \times d \times d$ parameters per layer. This is a non-trivial increase compared to the parameter-free RoPE. The TFLOPS comparison in Table 3 is against the entire model, which obscures the relative cost of the new positional encoding. A clearer analysis of the parameter and FLOP overhead of the attention block itself would be more transparent.
- The experiments are not sufficient. ViT-B is a small backbone in transformers and the performance gain can be potentially vanishing on large backbones like ViT-XL/DiT-XL.

**Questions:**

Could you provide a more direct quantification of the parameter and FLOPs overhead? Specifically, what is the percentage increase in parameters and computation for an attention block when switching from ROPE to HAROPE?

---

### Official Review · Reviewer_aRKm · 2025-11-01

**Soundness:** 2
**Presentation:** 3
**Contribution:** 2
**Rating:** 4
**Confidence:** 3

**Summary:**

The paper proposes a positional encoding scheme (HARoPE) that enhances the standard rotary positional encoding by introducing a learnable, head-wise linear transformation (via SVD) before the rotary mapping in the attention modules. They suggest that this design addresses limitations of rigid frequency allocation, axis-independence, and uniform head treatment in fine-grained image generation, resulting in improved structural bias modeling for image generation tasks.

**Strengths:**

- The paper conducts thorough ablation studies to evaluate the contribution of each component of the proposed method, including multi-head specialization, orthogonal transformations, and the SVD-based design.
- The authors present the background and related work clearly and coherently, making the technical context easy to follow and well-motivated.

**Weaknesses:**

- The claim regarding "misalignment with learned semantic subspaces" may be overstated. Since the model is trained end-to-end with these relative positional embeddings, it likely adapts its semantic subspaces accordingly.
- The claimed compositional improvements in text-to-image models are supported only by qualitative examples. Quantitative evaluation on established benchmarks such as T2I-CompBench would strengthen the paper’s evidence.
- The comparisons in Table 4 may not be entirely fair, as the models compared have differing numbers of trainable parameters—performance gains could stem partly from increased parameter count.
- The paper’s novelty is somewhat limited. The proposed SVD-based head-wise transformation essentially adds another linear mapping on top of the existing QKV projections. Given that attention already learns distinct projections for each head, the idea of head-specific transformations is not entirely new.

**Questions:**

- You mentioned MMDiT and FLUX as two different baselines. Could you clarify which specific model you are referring to as “MMDiT”? Since FLUX itself is an MMDiT-based architecture, it’s unclear what separate MMDiT model you are comparing against.

---

### Official Review · Reviewer_fYQT · 2025-11-01

**Soundness:** 3
**Presentation:** 3
**Contribution:** 2
**Rating:** 4
**Confidence:** 3

**Summary:**

This work introduces a head-wise adaptive rotary positional encoding (HARoPE) for image generation, which enables dynamic frequency reallocation, semantic alignment of rotary planes, and head-specific positional receptive fields. The proposed HARoPE generates better performance than RoPE.

**Strengths:**

- This work is well-organized.
- The motivation of the proposed HARoPE is clear and convincing. The proposed approach is simple and useful.
- The experimental results demonstrate its effectiveness.

**Weaknesses:**

This work claims that the proposed HARoPE enables dynamic frequency reallocation, semantic alignment of rotary planes, and head-specific positional receptive fields. However, the paper lacks detailed analysis or empirical evidence demonstrating that the introduced linear transformation layer indeed resolves these issues after training, which makes the argument for the method's effectiveness less convincing.

Moreover, this additional linear transformation introduces extra parameters during fine-tuning and can also be interpreted as a feature transformation applied to the query and key.

**Questions:**

- Does the network initialize and update $U_h$, $V_h$, and $\sum_h$ during training?

---

### Note · Authors · 2025-11-14

I have read and agree with the venue's withdrawal policy on behalf of myself and my co-authors.